# Hypergraph-Enhanced Contrastive Learning for Multi-View Clustering with Hyper-Laplacian Regularization

**Zhibin Gu**[1,2,3]  **Weili Wang**[4]*
[1]College of Computer and Cyber Security, Hebei Normal University, China
[2]Hebei Provincial Key Laboratory of Network & Information Security, Hebei Normal University, China
[3]Hebei Provincial Engineering Research Center for Supply Chain Big Data Analytics
& Data Security, Hebei Normal University, China
[4]Independent Researcher, China
guzhibin@hebtu.edu.cn, weiliw295@gmail.com

## Abstract

Deep multi-view clustering (DMVC) has emerged as a promising paradigm for integrating information from multiple views by leveraging the representation power of deep neural networks. However, most existing DMVC methods primarily focus on modeling pairwise relationships between samples, while neglecting higher-order structural dependencies among multiple samples, which may hinder further improvements in clustering performance. To address this limitation, we propose a hypergraph neural network (HGNN)-driven multi-view clustering framework, termed **H**ypergraph-enhanced c**O**ntrastive learning with hy**PE**r-Laplacian regula**R**ization (**HOPER**), a novel model that jointly captures high-order correlations and preserves local manifold structures across views. Specifically, we first construct view-specific hypergraph structures and employ the HGNN to learn node representations, thereby capturing high-order relationships among samples. Furthermore, we design a hypergraph-driven dual contrastive learning mechanism that integrates inter-view contrastive learning with intra-hyperedge contrastive learning, promoting cross-view consistency while maintaining discriminability within hyperedges. Finally, a hyper-Laplacian manifold regularization is introduced to preserve the local geometric structure within each view, thereby enhancing the structural fidelity and discriminative power of the learned representations. Extensive experiments on diverse datasets demonstrate the effectiveness of our approach.

## 1 Introduction

Multi-view data, collected from diverse sources or extracted via multiple feature extractors, contain both consensus and complementary information. As such data become increasingly prevalent in real-world applications, multi-view learning has emerged as a fundamental paradigm in machine learning for enhancing downstream performance by leveraging cross-view correlations [1–6]. Among its various tasks, multi-view clustering (MVC) plays a pivotal role in unsupervised learning by partitioning samples into meaningful groups without label supervision, thereby facilitating effective data analysis and organization [7–10] .

With the advance of deep representation learning, a variety of deep multi-view clustering (DMVC) methods have emerged, which can be broadly categorized into two main paradigms: representation

---

*Corresponding author

39th Conference on Neural Information Processing Systems (NeurIPS 2025).

learning-based and graph learning-based approaches [11]. The former typically leverages self-supervised learning frameworks to extract informative and discriminative latent representations directly from raw input features [12–14]. For instance, Wu et al. [15] employed view-specific deep autoencoders to extract embedded features and applied self-weighted contrastive fusion to learn robust global features. Cui et al. [16] enhanced information consistency across views and reduced redundancy by maximizing the lower bound of sufficient representation. Although deep representation learning methods have achieved significant success, they often struggle to explicitly model the complex relationships between samples, particularly those arising from intricate data structures. To address this limitation, deep graph learning methods have gained attention by using a shared graph neural network (GNN) encoder and projection head to represent each view as a graph, enabling unified cross-view representation learning in a common latent space [17–19]. For example, Xia et al. [20] employed graph convolutional networks (GCNs) to learn modality-specific representations, and introduced contrastive losses to encourage discriminative and clustering-friendly alignment across modalities. Similarly, Du et al. [21] utilized multiple GCNs with shared weights to extract view-specific representations, and incorporated a clustering embedding layer to jointly optimize representation learning and clustering performance. In addition, Dong et al. [22] performed contrastive learning at the graph structure level under the guidance of a consensus graph, thereby capturing the underlying structural information of the data.

Despite the performance improvements achieved by deep multi-view clustering (DMVC) methods through modeling pairwise relationships among samples, several critical limitations remain to be addressed. First, most existing DMVC methods primarily focus on modeling pairwise correlations between samples, while overlooking high-order and complex interactions among data points. This simplification significantly limits their ability to capture rich structural priors that are critical for effective clustering. Second, many contrastive learning-based DMVC approaches suffer from the false negative problem, where semantically similar samples from different views are incorrectly treated as negatives. This misidentification undermines the discriminative power of the learned representations. Third, these methods often lack explicit constraints to preserve high-order local geometric structures in the latent space, which are essential for maintaining topological consistency and enhancing the robustness of clustering.

To address the limitations of existing deep multi-view clustering methods, we propose a hypergraph-enhanced contrastive learning approach with hyper-Laplacian regularization. Specifically, for each view, we construct a hypergraph based on the sample features. This hypergraph structure, along with the corresponding features, is then input into a hypergraph neural network with shared weights to learn view-specific node representations, capturing high-order correlations among the data. Subsequently, a hypergraph-induced dual contrastive learning mechanism is employed to regularize the node representations: the inter-view contrastive loss regularization enhances consistency across views, while the intra-view contrastive loss regularization helps mitigate the false negative issue, promoting more discriminative representation learning. Finally, we introduce a hyper-Laplacian manifold regularization term to preserve the view-specific high-order local geometric structure, further enhancing the discriminative power of the node representations. In summary, the contributions of the proposed framework are as follows:

- We introduce a hypergraph neural network (HGNN)-based representation learning framework for multi-view clustering, which captures high-order data correlations by leveraging a hypergraph structure and a shared-weight HGNN.

- We propose a hypergraph-enhanced dual contrastive learning mechanism, which consists of an inter-view contrastive loss to reinforce consistency across different views and a intra-hyperedge contrastive loss to enhance the discriminability of individual samples within each hyperedge.

- We incorporate hyper-Laplacian manifold regularization to preserve view-specific higher-order local geometric structures, thereby further enhancing the robustness and effectiveness of representation learning.

- Experimental results demonstrate the superiority of our approach, highlighting its effectiveness in capturing high-order correlations and improving clustering performance compared to existing methods.

## 2 Related work

### 2.1 Hypergraph neural network

A hypergraph $\mathcal{G} = (\mathcal{V}, \mathcal{E})$ is a generalized graph structure capable of modeling high-order relationships, where $\mathcal{V}$ denotes the set of vertices and $\mathcal{E}$ denotes the set of hyperedges. The hyperedges can be represented using an incidence matrix $\mathbf{H} \in \{0, 1\}^{|\mathcal{V}| \times |\mathcal{E}|}$, where each entry $h(v, e)$ equals 1 if vertex $v$ belongs to hyperedge $e$, and 0 otherwise. Each hyperedge is associated with a non-negative weight, which can be encoded in a diagonal matrix $\mathbf{W} \in \mathbb{R}^{|\mathcal{E}| \times |\mathcal{E}|}$ with diagonal elements $w(e)$. The degree of a vertex and a hyperedge are defined as:

$$d(v) = \sum_{e \in \mathcal{E}} w(e)h(v, e), \quad d(e) = \sum_{v \in \mathcal{V}} h(v, e), \tag{1}$$

which can be organized into diagonal matrices $\mathbf{D}_v$ and $\mathbf{D}_e$, respectively.

Based on these definitions, the normalized hypergraph Laplacian matrix is formulated as:

$$\mathbf{L}_H = \mathbf{D}_v - \mathbf{HWD}_e^{-1}\mathbf{H}^\top \tag{2}$$

Owing to the expressive power of hypergraph structures in modeling high-order relationships, hypergraph neural networks (HGNNs) have attracted increasing attention in recent years. The general HGNN framework typically consists of two key components: hypergraph construction and hypergraph convolution [23]. According to whether hyperedge construction is performed explicitly or implicitly, hypergraph construction methods can be broadly categorized into four types: distance-based, representation-based, attribute-based, and network-based approaches [24]. Hypergraph convolutions can be further classified into spatial and spectral methods, depending on how the convolutional operators are defined [25]. Recently, hypergraph-based models have demonstrated impressive performance in clustering tasks. For instance, [26] and [27] leverage hypergraphs to effectively capture attribute information and high-order structural dependencies, achieving strong results in single-view node classification and clustering. However, most existing approaches are tailored to single-view settings, and relatively limited attention has been paid to exploring hypergraph learning in the context of multi-view clustering.

### 2.2 Multi-view contrastive clustering

In the domain of multi-view clustering, contrastive learning demonstrated strong potential in promoting cross-view alignment [28–33]. For instance, Trosten et al. [34] enhanced multi-view clustering by aligning representations at the instance level. Similarly, Xu et al. [35] introduced contrastive learning at multiple feature levels, including high-level semantic consistency and cluster-level consistency across views. Their approach effectively mitigated the negative impact of view-specific inconsistencies in low-level features, resulting in more stable representation alignment. Pan and Kang [36] proposed a method that first obtains a smooth node representation through graph filtering and then learns a robust consensus graph guided by graph contrastive loss for clustering. Xu et al. [37] introduced a self-supervised framework that utilizes global pseudo-labels to help different views collaboratively learn discriminative features, improving both the consistency and robustness of multi-view clustering. In addition, Zhang et al. [38] leveraged contrastive learning to align generated and real views by applying diffusion and reverse denoising processes to intra-view data, enabling the model to capture distributional consistency and improve clustering performance.

## 3 Method

In this section, we introduce the proposed HOPER model. We first provide an overview of the HOPER framework, then describe the latent representation learning based on hypergraph neural networks, the hypergraph-enhanced dual contrastive learning mechanism, and the hypergraph Laplacian regularization. Lastly, we present the unified loss function that integrates all these components.

### 3.1 Framework outline

Given a multi-view dataset $\{\mathbf{X}^v \in \mathbb{R}^{N \times D_v}\}_{v=1}^M$ with $N$ samples and $M$ views, where $\mathbf{X}^v$ denotes the raw input of the $v$-th view, the goal of multi-view clustering is to partition the data into $K$ clusters.

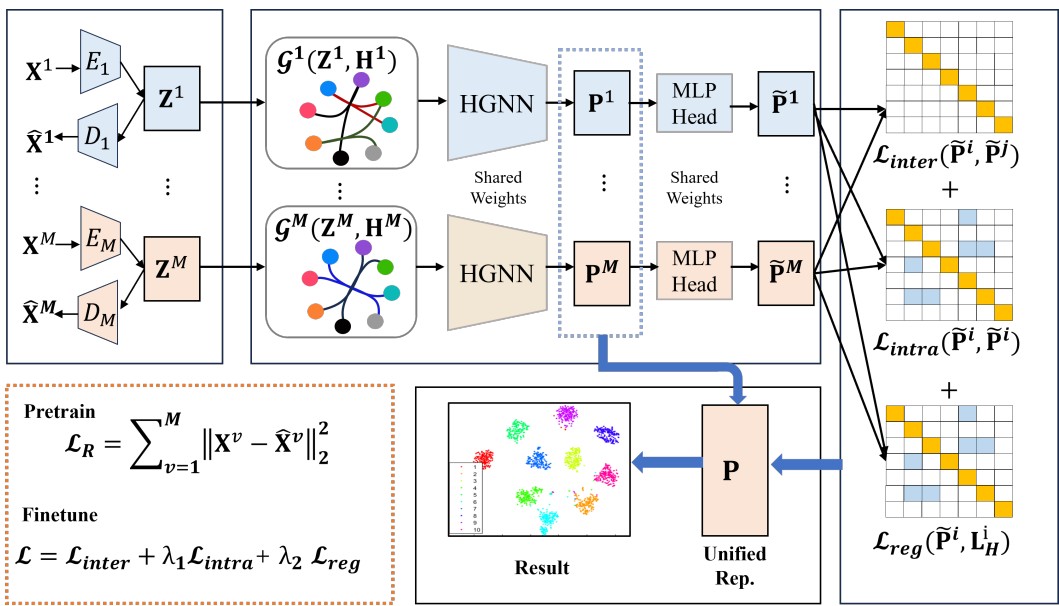

Figure 1: Illustration of the HOPER model. Given a multi-view dataset $\{\mathbf{X}^v\}_{v=1}^M$, we first learn latent representations and build view-specific hypergraphs. These are fed into a shared HGNN to generate embeddings. Then, a dual contrastive learning mechanism enhances consistency and discrimination. Finally, hyper-Laplacian regularization preserves the local geometric structure.

The overall framework of the proposed HOPER model is illustrated in Figure 1. Specifically, for each view, we first employ an autoencoder to project the raw features into a latent space and then construct a corresponding hypergraph structure based on the latent representations. The sample features and the corresponding hypergraph structure are then fed into a hypergraph neural network (HGNN) with shared parameters to learn node representations. Subsequently, a hypergraph-enhanced dual contrastive learning strategy is applied to the embeddings to improve both cross-view consistency and sample-level discriminability. Finally, a hyper-Laplacian regularization term is introduced to automatically preserve the local high-order geometric structure of each view in the embedding space.

### 3.2 Hypergraph construction and HGNN-based representation learning

To effectively model high-order correlations among data instances, we first construct view-specific hypergraph structures. Considering that raw multi-view data may contain noise and redundant information, directly constructing hyperedges based on the original features can degrade the quality of the hypergraphs. To address this issue, instead of relying on raw input features, we adopt an autoencoder to project the original data $\mathbf{X}^v$ into a compact latent space. Specifically, the autoencoder comprises an encoder $\mathbf{Z}^v = f(\mathbf{X}^v; \theta^v)$ and a decoder $\hat{\mathbf{X}}^v = g(\mathbf{Z}^v; \phi^v)$, where $\theta^v$ and $\phi^v$ are trainable parameters. The model is trained to minimize the reconstruction loss, as defined in Eq. (3).

$$\mathcal{L}_R = \sum_{v=1}^M \|\mathbf{X}^v - \hat{\mathbf{X}}^v\|_2^2 \tag{3}$$

After training, compact latent embeddings $\mathbf{Z}^v$ are obtained from the encoder module. Subsequently, multi-view hypergraph structures are constructed by applying a $k$-nearest neighbors strategy to these latent embeddings, where $k$ is a predefined parameter specifying the number of neighbors to select. Specifically, for each view, we compute the pairwise Euclidean distances between all samples to capture local geometric relationships. For each sample, we identify its top-$k$ nearest neighbors and construct a hyperedge connecting the sample with its neighbors. This process generates a collection of hyperedges that encode high-order relationships beyond simple pairwise connections, represented by the incidence matrix $\mathbf{H}^v$. Finally, the hypergraph for the $v$-th view is obtained, denoted as $\mathcal{G}^v = (\mathcal{V}^v, \mathcal{E}^v)$, with feature matrix $\mathbf{Z}^v$, and incidence matrix $\mathbf{H}^v$.

To further enhance the discriminative capability of latent representations by leveraging high-order relationships among samples, we incorporate a hypergraph neural network (HGNN) into our framework to learn more expressive node embeddings. Specifically, we adopt a two-layer HGNN module composed of stacked hypergraph convolution layers for node representation learning. Following [23], a hyperedge convolutional layer is formulated as:

$$\mathbf{P}_l^v = \sigma(\mathbf{D}_v^{-1}\mathbf{H}\mathbf{W}\mathbf{D}_e^{-1}\mathbf{H}^\top\mathbf{P}_{l-1}^v\Theta_l), \tag{4}$$

where $\sigma$ denotes the nonlinear activation function, $\Theta_l$ denotes the learnable parameter, $\mathbf{P}_l^v$ is the node representation at $l$ layer, $\mathbf{P}_0^v = \mathbf{Z}^v$. Notably, the HGNN network of different views share parameters in order to better align the learned representations across views.

## 3.3 Hypergraph-enhanced dual contrastive learning

Multi-view data provide complementary perspectives of the same underlying object, and typically exhibit semantic consistency across views. This cross-view consistency can be effectively exploited through contrastive learning. However, existing multi-view contrastive learning methods focus on aligning paired samples across views, treating them as positives while largely ignoring intra-view structural characteristics. Specifically, in each view, samples connected by the same hyperedge in the underlying hypergraph often interact through a message passing scheme, which can induce an over-smoothing effect—making originally distinguishable samples overly similar in representation space. While such samples may share local semantic information, they should still remain discriminative to preserve meaningful structural distinctions. The failure to account for this nuance limits the expressiveness of the learned representations, ultimately hindering clustering performance.

To address this limitation, we propose a hypergraph-enhanced dual contrastive learning mechanism, which jointly performs inter-view and intra-view contrastive learning. Specifically, inter-view contrastive learning aligns representations across views by encouraging cross-view consistency, while the intra-view contrastive learning leverages hyperedge structural information to enhance the discriminability among samples within each view.

**Inter-view contrastive loss:** This loss function is typically designed to compare node representations learned from different views, aiming to maintain consistency across views. Let $\tilde{\mathbf{P}}^v$ denote the node features, i.e., the output of the projection head for node representations. Specifically, representations of the same instance across different views are treated as positive pairs, while all other instances are regarded as negative pairs. The inter-view contrastive loss $\mathcal{L}_{inter}$ is defined as:

$$\mathcal{L}_{inter}(\tilde{\mathbf{p}}_i^v) = -\log\frac{\exp\left(s(\tilde{\mathbf{p}}_i^v, \tilde{\mathbf{p}}_i^m)/\tau\right)}{\sum_{j=1}^N\sum_{m\neq v}\exp\left(s(\tilde{\mathbf{p}}_i^v, \tilde{\mathbf{p}}_j^m)/\tau\right)}, \tag{5}$$

where $\tau$ denotes the temperature parameter, $s(\cdot)$ denotes the similarity function which is implemented as cosine similarity.

**Intra-view contrastive loss:** Due to the strong connectivity introduced by hyperedges, the message passing mechanism in HGNN may lead to an over-smoothing issue, where node representations tend to become indistinguishable—especially for nodes connected by the same hyperedge—thus undermining their discriminative power [39]. To mitigate this, we introduce an intra-view contrastive learning strategy that enhances the distinctiveness of individual samples. Specifically, each node is treated as a positive pair with itself, while other nodes within the same hyperedge are considered negative samples. The intra-view contrastive loss $\mathcal{L}_{intra}$ is formulated as

$$\mathcal{L}_{\text{intra}}(\tilde{\mathbf{p}}_i^v) = -\log\frac{\exp\left(s(\tilde{\mathbf{p}}_i^v, \tilde{\mathbf{p}}_i^v)/\tau\right)}{\sum_{j\in\mathcal{N}_i^v}\exp\left(s(\tilde{\mathbf{p}}_i^v, \tilde{\mathbf{p}}_j^v)/\tau\right)}, \tag{6}$$

where $\mathcal{N}_i^v$ denotes the set of neighbors which are in the same hyperedges as node $i$ in $v$-th view. In our experiments, the temperature parameters in Equation (5) and Equation (6) are shared optimized.

By integrating the aforementioned dual contrastive learning strategy, the model not only enforces global alignment across views but also enhances local discriminability within each view, thereby yielding more robust and semantically meaningful representations.

## 3.4 Hyper-Laplacian regularization

To better preserve the intrinsic local structure of the data, we incorporate a hypergraph Laplacian regularization term into our model. By leveraging the hypergraph Laplacian, this regularizer embeds

**Algorithm 1** Hypergraph-enhanced Multi-view Representation Learning

---

**Input:** Multi-view raw features $\{\mathbf{X}^v\}_{v=1}^M$, number of clusters $K$
**Output:** Cluster assignments via $k$-means on unified representations
1: **Pretraining:**
2: **for** each view $v = 1$ to $M$ **do**
3:     Pretrain the view-specific autoencoder by optimizing Eq.(3)
4: **end for**
5: **Feature Encoding:**
6: Obtain node features $\{\mathbf{Z}^v\}_{v=1}^M$ from encoder networks
7: **Hypergraph Construction:**
8: Construct view-specific hypergraphs via $k$-NN strategy on $\mathbf{Z}^v$
9: **Joint Optimization:**
10: **for** $t = 1$ to $T_{\max}$ **do**
11:     Update the shared hypergraph encoder and projection head by optimizing Eq.(8)
12: **end for**
13: **Fusion:**
14: Compute unified representations using Eq.(9)
15: **Clustering:**
16: Apply $k$-means to obtain final clustering results

---

the manifold assumption into the learning process—that is, if multiple data points are close in the intrinsic geometry of the data space, their representations in the latent space should also be similar. This mechanism promotes smoothness of the learned representations over the hypergraph, thereby enhancing the model's ability to capture higher-order structural information and improving its generalization performance on downstream tasks. The mathematical expression is as follows:

$$\mathcal{L}_{reg} = \sum_{v=1}^M tr(\tilde{\mathbf{P}}^v \mathbf{L}_h^v (\tilde{\mathbf{P}}^v)^\top), \tag{7}$$

where $\mathbf{L}_h^v$ is hyper-graph Laplacian matrix of the $v$-th view. By introducing this hyper-Laplacian regularization, our model gains the ability to exploit richer structural information beyond simple pairwise constraints, leading to improved representation quality and better generalization performance across downstream tasks.

### 3.5 The overall loss function

By integrating the intra-view and inter-view contrastive loss, and the hypergraph Laplacian regularization, the overall loss function of the proposed HOPER model is formulated as follows:

$$\mathcal{L} = \mathcal{L}_{inter} + \lambda_1 \mathcal{L}_{intra} + \lambda_2 \mathcal{L}_{reg} \tag{8}$$

Overall, the optimization of our method consists of two stages: initialization and fine-tuning. During the initialization stage, we first pretrain a view-specific autoencoder for each view by minimizing the reconstruction loss, as defined in Eq.(3). Once pretraining is completed, we construct a hypergraph structure for each view based on the learned latent representations. In the fine-tuning stage, the entire network is trained by optimizing the objective function given in Eq.(8). After optimization, the learned node embeddings from all views are concatenated to form the unified representation $\mathbf{P}$:

$$\mathbf{P} = [\mathbf{P}^1, \mathbf{P}^2, ..., \mathbf{P}^M] \in \mathbb{R}^{N \times \sum_{v=1}^M d_v}, \tag{9}$$

where $d_v$ denotes the dimension of $\mathbf{P}^v$ Finally, the unified representation $\mathbf{P}$ is used as input to the $k$-means algorithm to produce the clustering results. The whole learning process is summarized in Algorithm 1.

### 3.6 Comparison with Previous Studies

Although recent studies have explored Hypergraph Contrastive Learning (HCL) [40–44] and Hyper-Laplacian Regularization [45–51], our approach differs fundamentally from these existing methods.

First, regarding application scenarios, existing HCL methods [40–42] generally treat multi-view data as augmented variants of a single view, whereas our framework defines multi-view data as heterogeneous feature sets extracted from the same instance, capturing genuinely distinct and complementary perspectives. Second, the learning objectives differ significantly: [40–42] primarily target node classification, while [43] focuses on recommendation tasks. In contrast, our model is specifically designed for unsupervised multi-view clustering, aiming to discover shared semantics and cross-view consistency without label supervision. Although [44] incorporates HCL into multi-view clustering, the two methods diverge fundamentally in terms of model design motivation, the perspective for exploiting multi-view data, and the contrastive learning mechanism. Furthermore, while some studies have incorporated Hyper-Laplacian Regularization into multi-view clustering [45–52], most adopt shallow learning paradigms based on matrix/tensor factorization or graph self-representation, lacking the representational capacity of deep neural models. In contrast, our approach leverages a deep learning framework that integrates Hyper-Laplacian Regularization with hypergraph neural networks, enabling richer feature representations and improved clustering quality.

## 4 Experiments

This section presents a comprehensive empirical evaluation of the proposed HOPER model, encompassing experimental settings, performance comparisons, parameter sensitivity analysis, feature visualizations, and ablation studies.

### 4.1 Experimental settings

**Datasets:** To comprehensively evaluate the effectiveness of the proposed HOPER framework, we conduct experiments on six publicly available multi-view datasets. Their statistical details are shown in Table 1. BBCsport contains 544 samples with 2 views, corresponding to five categories. Synthetic3d is a 3-D dataset containing 600 samples with 3 views. WebKB contains 203 web pages of 4 categories. Each web page is described from 3 views. COIL-20 consists of 1440 samples with 3 views which belongs to 20 categories. Handwritten contains 2000 samples of handwritten digits from 0-9, where each sample is described from 6 views. Hdigit is a digit dataset from MNIST Handwritten Digits and USPS Handwritten Digits which consists of 10000 samples described by 2 views.

Table 1: Statistics of six benchmark datasets.

| Dataset | #Samples | #Views | #Clusters |
|---------|----------|--------|-----------|
| BBCSport | 544 | 2 | 5 |
| Synthetic3d | 600 | 3 | 3 |
| WebKB | 1051 | 2 | 2 |
| COIL-20 | 1440 | 3 | 20 |
| Handwritten | 2000 | 6 | 10 |
| Hdigit | 10000 | 2 | 10 |

**Evaluation metrics:** For evaluation, three widely-used metrics, including the clustering Accuracy (ACC), Normalized Mutual Information (NMI), Adjusted Rand Index (ARI) are calculated to comprehensively compare the performance of various methods.

**Comparison methods:** We compare our framework with the following state-of-the-art DMVC algorithms to investigate the effectiveness of our framework, i.e., MFLVC (2022) [35], CVCL (2023) [53], DealMVC (2023) [54], SEM (2023) [55], DIVIDE (2024) [56], MAGA (2024) [33].

**Implementation details:** The proposed framework consists of two main modules: initialization and fine-tuning. In the initialization module, we utilize a four-layer autoencoder to obtain the latent embeddings. The number of nearest neighbors of hyperedge construction is tuned over $\{5, 10, 15, 20, 25, 30\}$ on different datasets. In the fine-tuning module, we optimize the hyperparameters $\lambda_1$ and $\lambda_2$. Based on empirical observations, we perform a grid search over the values $\{0.0001, 0.001, 0.01, 0.1, 1, 5, 10\}$ to select the optimal values for both hyperparameters on multiple datasets. The best performance is obtained under the combination of $\lambda_1 = 5$ and $\lambda_2 = 0.0001$. These specific values were then used for all experiments reported in this paper. The training process consists of 2000 epochs for the autoencoder initialization phase and 200 epochs for the fine-tuning phase. We employ a cosine learning rate decay to adjust the learning rate dynamically. All experiments are conducted using the PyTorch framework on an NVIDIA GeForce RTX 3090 GPU.

Table 2: Clustering performance across benchmark datasets.

| Dataset | Metric | MFLVC | CVCL | DealMVC | SEM | DIVIDE | MAGA | HOPER |
|---|---|---|---|---|---|---|---|---|
| BBCSport | ACC | 0.7224 | 0.6211 | 0.8070 | 0.6085 | 0.4467 | 0.5533 | **0.9504** |
| | NMI | 0.5344 | 0.3645 | 0.6559 | 0.3666 | 0.1507 | 0.2808 | **0.8509** |
| | ARI | 0.5874 | 0.3137 | 0.6005 | 0.2918 | 0.1091 | 0.2286 | **0.8677** |
| Synthetic3d | ACC | 0.9500 | 0.9546 | 0.8033 | 0.9467 | 0.9497 | 0.9600 | **0.9717** |
| | NMI | 0.8218 | 0.8158 | 0.5797 | 0.8095 | 0.8083 | 0.8388 | **0.8775** |
| | ARI | 0.8582 | 0.8689 | 0.5667 | 0.8494 | 0.8553 | 0.8846 | **0.9165** |
| WebKB | ACC | 0.7174 | 0.7181 | 0.6974 | 0.9486 | 0.8325 | 0.9010 | **0.9829** |
| | NMI | 0.2986 | 0.2832 | 0.2474 | 0.6809 | 0.1791 | 0.4201 | **0.8416** |
| | ARI | 0.1885 | 0.7810 | 0.1552 | 0.7897 | 0.2985 | 0.5986 | **0.9249** |
| COIL-20 | ACC | 0.3875 | 0.6882 | 0.2299 | 0.7403 | 0.6486 | 0.4569 | **0.8285** |
| | NMI | 0.5450 | 0.7851 | 0.4783 | 0.8296 | 0.7608 | 0.6317 | **0.8880** |
| | ARI | 0.3093 | 0.7007 | 0.1735 | 0.6588 | 0.5466 | 0.4151 | **0.7869** |
| Handwritten | ACC | 0.8990 | 0.9194 | 0.8220 | 0.7645 | 0.8708 | 0.9415 | **0.9685** |
| | NMI | 0.8259 | 0.8878 | 0.8163 | 0.7285 | 0.8277 | 0.9083 | **0.9317** |
| | ARI | 0.7939 | 0.8473 | 0.7367 | 0.6317 | 0.8086 | 0.8854 | **0.9308** |
| Hdigit | ACC | 0.9882 | 0.9505 | **0.9980** | 0.9864 | 0.9646 | 0.9954 | 0.9961 |
| | NMI | 0.9841 | 0.8900 | **0.9934** | 0.9606 | 0.9103 | 0.9856 | 0.9876 |
| | ARI | 0.9882 | 0.8930 | **0.9980** | 0.9701 | 0.9229 | 0.9898 | 0.9961 |

## 4.2 Performance comparison

Table 2 summarizes the performance of all methods, with the best and second-best results highlighted in **bold** and underlined, respectively. The experimental comparison demonstrates that the HOPER model achieves highly competitive clustering performance compared to existing baseline methods across all evaluation metrics. For instance, on the COIL-20 dataset, HOPER improves upon the second-best SEM model by approximately 8.8%, 5.4%, and 12.8% in terms of ACC, NMI, and ARI, respectively. These results validate that our approach, leveraging hypergraph-enhanced contrastive learning and hypergraph Laplacian manifold regularization, enhances the discriminability of latent representations, thereby improving clustering performance. Moreover, although the HOPER model does not achieve the best result on the Hdigit dataset, its accuracy is nearly 100% and only 0.19% lower than that of DealMVC, further validating the effectiveness of our approach.

## 4.3 Parameter sensitivity analysis

In the HOPER model, two hyperparameters, $\lambda_1$ and $\lambda_2$, are introduced to balance the contrastive learning objective and the hyper-Laplacian regularization term. To evaluate the model's sensitivity, we conduct a grid search over $\lambda_1 \in \{1, 2, 3, 4, 5\}$ and $\lambda_2 \in \{0.0001, 0.0002, 0.0003, 0.0004, 0.0005\}$. The clustering results (ACC) on various datasets are reported in Figure 2. On Synthetic3D, WebKB, and COIL-20 datasets, HOPER demonstrates consistent and stable clustering performance across the range of parameter values. Some fluctuations are observed on BBCSport, Handwritten, and Hdigit datasets, which may be attributed to the trade-off between the regularization terms affecting the discriminative quality of the learned representations. Overall, despite minor variations, HOPER maintains robust and competitive clustering results within the evaluated hyperparameter ranges.

## 4.4 Visualization of representation evolution

In this section, we present a qualitative analysis of the proposed HOPER model on the BBCSport dataset by visualizing the learned representations at different stages of the framework. As shown in Figure 3, subfigures (a) and (b) illustrate the raw data from two views, $\mathbf{X}^1$ and $\mathbf{X}^2$, while (c) and (d) show the corresponding latent embeddings $\mathbf{Z}^1$ and $\mathbf{Z}^2$ obtained via the autoencoder module. Compared with the raw inputs, the latent embeddings reveal a clearer clustering structure, suggesting that the autoencoder effectively denoises the data and captures more compact representations. Subfigures (e) and (f) visualize the node representations $\mathbf{P}^1$ and $\mathbf{P}^2$ refined by the hypergraph contrastive learning module and the hypergraph Laplacian regularization. These representations exhibit more discriminative and well-separated clusters than their latent counterparts, highlighting the effectiveness of the proposed hypergraph-based enhancements in capturing high-order and local relational structures

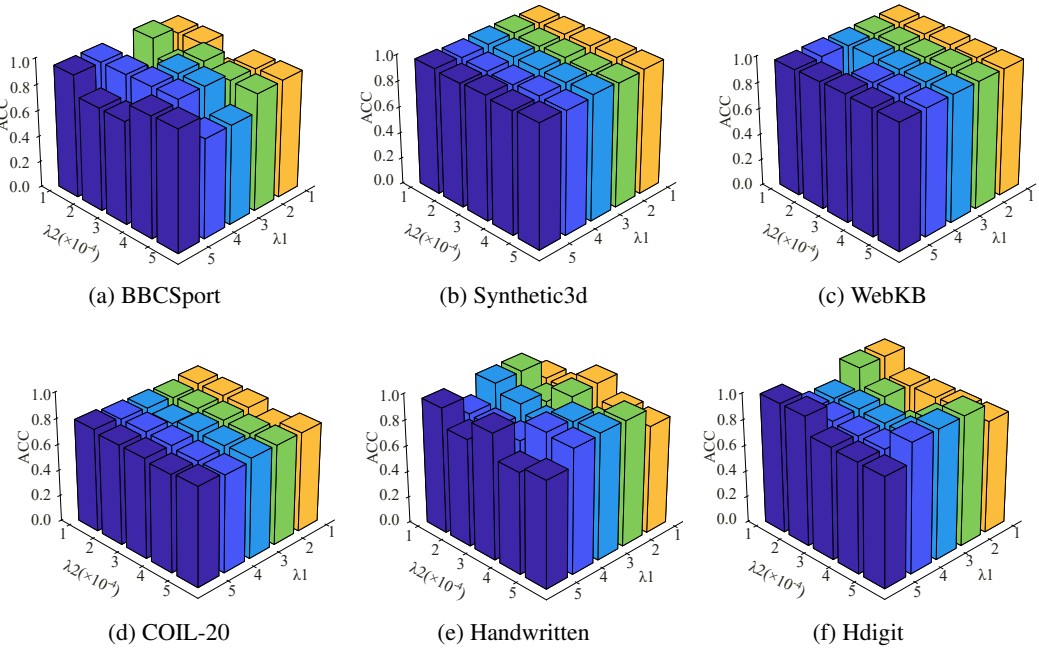

(a) BBCSport     (b) Synthetic3d     (c) WebKB

(d) COIL-20     (e) Handwritten     (f) Hdigit

Figure 2: Hyperparameter sensitivity analysis of the HOPER model on multiple datasets.

within each view. Finally, subfigure (g) depicts the unified representation aggregated from multiple views. It exhibits the most compact and separable cluster structures among all stages, demonstrating the ability of HOPER to effectively exploit cross-view complementary information and enhance the overall representation quality, thereby leading to improved clustering performance.

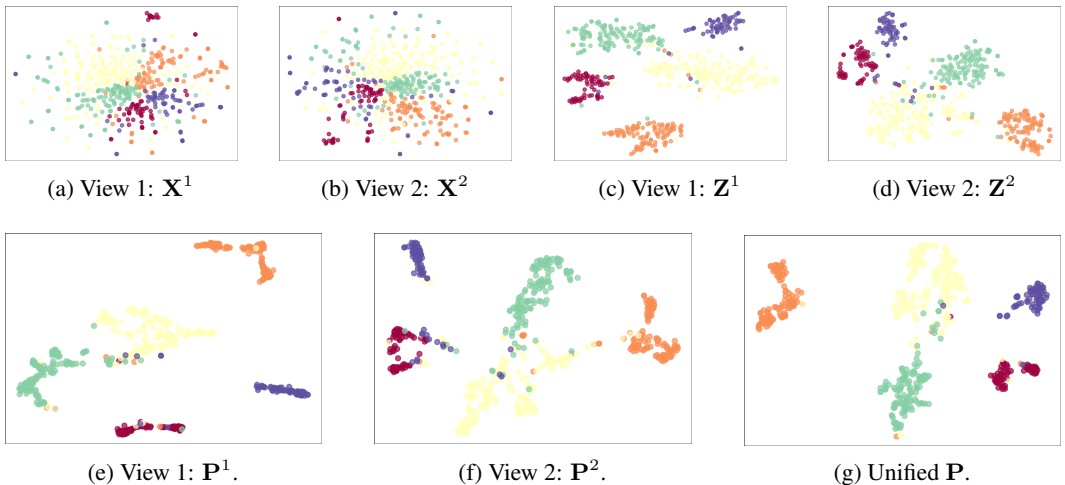

(a) View 1: $\mathbf{X}^1$    (b) View 2: $\mathbf{X}^2$    (c) View 1: $\mathbf{Z}^1$    (d) View 2: $\mathbf{Z}^2$

(e) View 1: $\mathbf{P}^1$.    (f) View 2: $\mathbf{P}^2$.    (g) Unified $\mathbf{P}$.

Figure 3: Visualization of the learned representations at different stages of the HOPER.

## 4.5 Convergence analysis

In this subsection, we demonstrate the convergence of HOPER across six datasets by reporting the loss values. As shown in Figure 4, the horizontal axis represents the training epochs and the vertical axis denotes the loss value. It can be observed that the loss drops rapidly during the first 100 epochs and then gradually decreases until convergence.

### 4.6 Ablation studies

As defined in Eq. 8, the overall loss function of HOPER consists of three components. To investigate the individual contribution of each term, we conduct ablation studies by systematically removing each component and retraining the model under the same experimental settings. Table 3 reports the clustering performance on multiple datasets under different loss configurations, where a checkmark indicates that the corresponding loss term is included. The experimental results demonstrate that removing any single component from the complete HOPER model consistently leads to performance degradation across all datasets, with some cases exhibiting significant drops. This highlights that HOPER effectively integrates inter-view contrastive learning, intra-view contrastive learning, and hypergraph Laplacian manifold regularization, which together promote cross-view consistency in sample representations while preserving inter-sample discriminability, ultimately enhancing clustering performance.

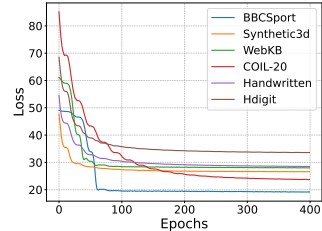

Figure 4: Convergence analysis of the HOPER model on multiple datasets.

Table 3: Ablation study on the effects of individual loss components in HOPER.

| Components | | | BBCSport | | | Synthetic3d | | | WebKB | | |
|---|---|---|---|---|---|---|---|---|---|---|---|
| $\mathcal{L}_{intra}$ | $\mathcal{L}_{inter}$ | $\mathcal{L}_{reg}$ | ACC | NMI | ARI | ACC | NMI | ARI | ACC | NMI | ARI |
| ✓ | ✓ | | 0.8897 | 0.7785 | 0.8076 | 0.9650 | 0.8624 | 0.8990 | 0.9686 | 0.7520 | 0.8659 |
| ✓ | | ✓ | 0.9449 | 0.8371 | 0.8529 | 0.9500 | 0.8117 | 0.9504 | 0.9724 | 0.7744 | 0.8815 |
| | ✓ | ✓ | 0.5147 | 0.4724 | 0.2553 | 0.9667 | 0.8607 | 0.9027 | 0.9629 | 0.7151 | 0.8387 |
| ✓ | ✓ | ✓ | **0.9504** | **0.8509** | **0.8677** | **0.9717** | **0.8775** | **0.9165** | **0.9829** | **0.8416** | **0.9249** |
| Components | | | COIL-20 | | | Handwritten | | | Hdigit | | |
| $\mathcal{L}_{intra}$ | $\mathcal{L}_{inter}$ | $\mathcal{L}_{reg}$ | ACC | NMI | ARI | ACC | NMI | ARI | ACC | NMI | ARI |
| ✓ | ✓ | | 0.7910 | 0.8740 | 0.7597 | 0.8285 | 0.8573 | 0.7857 | 0.7022 | 0.8661 | 0.7173 |
| ✓ | | ✓ | 0.7549 | 0.8781 | 0.7358 | 0.8695 | 0.8529 | 0.7915 | 0.8471 | 0.9117 | 0.8406 |
| | ✓ | ✓ | 0.7847 | 0.8687 | 0.7496 | 0.8240 | 0.8814 | 0.8410 | 0.9268 | 0.8439 | 0.6537 |
| ✓ | ✓ | ✓ | **0.8285** | **0.8880** | **0.7869** | **0.9685** | **0.9317** | **0.9308** | **0.9961** | **0.9876** | **0.9961** |

## 5 Conclusion

This paper proposes a novel multi-view clustering framework, HOPER, which integrates hypergraph-enhanced contrastive learning with hypergraph Laplacian regularization to learn discriminative feature representations. Specifically, HOPER captures high-order relationships among samples through hypergraph construction and hypergraph neural networks. To further improve representation quality, a hypergraph-driven dual contrastive learning mechanism is introduced, comprising inter-view contrastive learning and intra-hyperedge contrastive learning, which promotes cross-view consistency while preserving discriminability within hyperedge. In addition, hypergraph Laplacian regularization is employed to preserve high-order local structural information. Extensive experiments on six benchmark datasets demonstrate that HOPER achieves highly competitive performance, validating its effectiveness for discriminative representation learning.

## 6 Limitations

A potential limitation of our method is its relatively higher computational complexity compared to traditional graph-based approaches. This is because hypergraphs introduce hyperedges that connect multiple nodes to capture high-order relationships, leading to more complex operations involving the incidence and Laplacian matrices than in conventional graphs where edges link only two nodes.

## Acknowledgements

This work was supported by the Natural Science Foundation of Hebei Province (No. F2025205006), the Science Foundation of Hebei Normal University (No. L2025B38) and the Backbone Talent Program (Program for Returned Overseas Scholars) (No. A2025016).

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
