# OpenReview forum: "Hypergraph-Enhanced Contrastive Learning for Multi-View Clustering with Hyper-Laplacian Regularization"
_NeurIPS.cc/2025/Conference — NeurIPS 2025 poster_

### Official Review · Reviewer_S36h · 2025-06-27

**Clarity:** 3
**Significance:** 4
**Originality:** 3
**Rating:** 4
**Confidence:** 4

**Summary:**

This work proposes a hypergraph neural network-based framework for deep multi-view clustering. The method addresses the limitation of existing approaches that focus primarily on pairwise relationships by introducing view-specific hypergraphs to model high-order sample dependencies. It further incorporates a dual contrastive learning strategy to enhance both cross-view consistency and intra-hyperedge discriminability. In addition, a hyper-Laplacian regularization term is employed to preserve local geometric structures within each view. Together, these components contribute to improved clustering performance through more comprehensive structural representation.

**Questions:**

See Weaknesses.

**Ethical Concerns:**

["NO or VERY MINOR ethics concerns only"]

**Final Justification:**

In the rebuttal, the authors' response has addressed my concerns.  I would like to keep my original positive score.

**Limitations:**

No discussion of limitations or societal risks is provided. It would be beneficial for the authors to acknowledge any potential drawbacks or broader consequences of the proposed method.

**Paper Formatting Concerns:**

The paper is well-formatted and consistent with the NeurIPS 2025 submission standards.

**Quality:**

3

**Strengths And Weaknesses:**

Strengths：
1. The paper introduces view-specific hypergraphs to capture high-order sample relationships, addressing limitations of pairwise multi-view clustering.
2. A novel inter-view and intra-hyperedge contrastive learning framework enhances cross-view consistency and within-hyperedge discriminability.
3. Experiments on multiple benchmarks show superior performance over recent methods, supported by visualizations that aid interpretability.

Weaknesses:
1. The introduction lacks adequate clarification of some necessary concepts, making certain sections difficult to understand. For instance, when the authors mention using hypergraphs to model high-order correlations, it is unclear whether these high-order correlations refer to inter-view or intra-view relationships.
2. Section 3.3 lacks a clear explanation of the motivation behind the Hypergraph-enhanced Dual Contrastive Learning. The phrase “treating them as positives” is ambiguous, as it is unclear what “them” specifically refers to. Clarifying this point would help improve understanding.
3. The single line of ``Input'' in the pseudocode is insufficient, as it omits the hyperparameters \(\lambda_1\) and \(\lambda_2\).
4. Some experimental settings are unclear. For example, the parameter configurations in Section 4.1 (“Implementation details”) are inconsistent with the hyperparameter values reported in Section 4.3, and no explanation is provided for this discrepancy.

---

> ### Author Rebuttal · Authors · 2025-07-30
>
> ** Weakness 1: The introduction lacks adequate clarification of some necessary concepts, making certain sections difficult to understand. For instance, when the authors mention using hypergraphs to model high-order correlations, it is unclear whether these high-order correlations refer to inter-view or intra-view relationships.**
>
> **A1:** Thank you for your constructive comment. In our method, the high-order correlations captured by the hypergraph primarily refer to intra-view structural relationships among samples within the same view. These correlations are modeled by connecting multiple similar samples through shared hyperedges, which reflect complex local structures beyond pairwise interactions. In contrast, inter-view consistency is addressed through our proposed inter-view contrastive learning mechanism. We will clarify this distinction more explicitly in the revised manuscript to improve overall conceptual clarity.
>
> **Weakness 2: Section 3.3 lacks a clear explanation of the motivation behind the Hypergraph-enhanced Dual Contrastive Learning. The phrase “treating them as positives” is ambiguous, as it is unclear what “them” specifically refers to. Clarifying this point would help improve understanding.**
>
> **A2: ** Thank you for your helpful feedback. The phrase “treating them as positives” specifically refers to sample pairs representing the same instance across different views in the inter-view contrastive learning component. These pairs are treated as positive examples, while all other instances are considered negatives. The intra-view contrastive learning component, on the other hand, treats each sample as a positive with itself and treats its neighbors within the same hyperedge as negatives, in order to mitigate the over-smoothing issue caused by message passing in hypergraph neural networks. We will revise this section to clearly distinguish these elements and better explain the underlying motivation in the revised manuscript.
>
> **Weakness 3: The single line of ``Input'' in the pseudocode is insufficient, as it omits the hyperparameters  $\lambda_1$ and  $\lambda_2$.**
>
> **A3: **  Thank you for your careful observation. We acknowledge that the current “Input” line in the pseudocode is incomplete and omits important hyperparameters $\lambda_1$ and $\lambda_2$. In the revised manuscript, we will update the pseudocode to explicitly include all relevant hyperparameters in the input specification to ensure completeness and clarity.
>
>
> **Weakness 4: Some experimental settings are unclear. For example, the parameter configurations in Section 4.1 (“Implementation details”) are inconsistent with the hyperparameter values reported in Section 4.3, and no explanation is provided for this discrepancy.**
>
> **A4:**  Thank you very much for your valuable comment. The inconsistency between the parameter configurations described in Section 4.1 and the hyperparameter values reported in Section 4.3 was due to a typographical oversight on our part. In fact, the hyperparameters follow the settings described in Section 4.3, where $\lambda_1$  is chosen from the range
> [1,2,3,4,5] and $\lambda_2$ from [0.0001,0.0002,0.0003,0.0004,0.0005]. We will clarify this point in the revised manuscript to ensure transparency and consistency throughout the paper.
>
> **Limitations: No discussion of limitations or societal risks is provided. It would be beneficial for the authors to acknowledge any potential drawbacks or broader consequences of the proposed method.**
>
> **A5:**  Thank you for your valuable suggestion. One potential limitation of our method is the increased computational complexity introduced by hypergraph modeling. This is mainly due to the need to capture high-order relationships through hyperedges connecting multiple nodes simultaneously, which results in larger and denser incidence matrices and more complex message-passing operations. In the revised manuscript, we will provide a more detailed discussion of this limitation.

---

### Official Review · Reviewer_4ypD · 2025-06-28

**Clarity:** 2
**Significance:** 3
**Originality:** 2
**Rating:** 5
**Confidence:** 4

**Summary:**

The manuscript addresses a key limitation of current deep multi-view clustering methods that overlook higher-order sample interactions by focusing only on pairwise relations. To tackle this, the authors leverage hypergraph neural networks combined with a dual contrastive learning strategy to improve cross-view alignment and intra-hyperedge discrimination. The integration of hyper-Laplacian regularization further preserves local manifold structures. Experimental results on multiple benchmarks validate the method’s superior clustering performance.

**Questions:**

(1) The definitions and roles of inter-view and intra-view contrastive losses are not clearly articulated. It remains unclear how each specifically contributes to cross-view consistency and within-view discriminability.

(2) Equation (7) lacks a detailed explanation of the construction of the hypergraph Laplacian matrix and the mechanism behind the hyper-Laplacian regularization term.

(3) Figure 2 indicates that the proposed method exhibits stable performance on some datasets but fluctuates significantly on others. The manuscript should discuss potential reasons for this variation in robustness.

(4) While hypergraph construction is intended to capture high-order relations to enhance representation learning, the underlying mechanism is not sufficiently explained and would benefit from a more in-depth discussion.

**Ethical Concerns:**

["NO or VERY MINOR ethics concerns only"]

**Final Justification:**

This paper proposed a simple but effective method, and I have no further concerns.

**Limitations:**

A potential limitation of hypergraph modeling is the increased computational complexity associated with capturing high-order relationships. This aspect should be explicitly acknowledged and discussed in the manuscript.

**Paper Formatting Concerns:**

No major formatting issues noted.

**Quality:**

3

**Strengths And Weaknesses:**

Strengths：

(1) The manuscript is well-structured, and the overall logic is rigorous and coherent.

(2) The use of high-order structural priors and contrastive learning to facilitate clustering-friendly representation learning is a novel and interesting idea.

(3) The experimental setup is clearly presented, and the proposed method achieves better performance than recent multi-view clustering approaches.

Weaknesses:

(1) The description of the dual contrastive losses is insufficient and requires further clarification.

(2) The descriptions of some equations are insufficient; specifically, the temperature parameters in Equations (5) and (6) lack clarity regarding whether they are shared or separately optimized.

(3) There are formatting inconsistencies, including mixed italic and upright subscripts in Equations (5) and (6), as well as inconsistent table header styles.

---

> ### Author Rebuttal · Authors · 2025-07-30
>
> **Weakness 1: The description of the dual contrastive losses is insufficient and requires further clarification.**
>
> **A1:**   Thank you for your valuable feedback. We agree that the original explanation of the dual contrastive losses could benefit from further clarification. To clarify, our proposed hypergraph-enhanced dual contrastive learning jointly optimizes two complementary objectives. The inter-view contrastive loss aligns representations of the same instance across different views, enforcing global semantic consistency. Meanwhile, the intra-view contrastive loss addresses the over-smoothing issue caused by message passing within hypergraphs by treating each node as a positive pair with itself and considering other nodes within the same hyperedge as negatives. This promotes local discriminability and preserves meaningful structural differences within each view. By combining these two losses, our model effectively captures both cross-view alignment and fine-grained intra-view distinctions, leading to more robust and semantically rich representations. We will further elaborate on this point in the revised manuscript.
>
> **Weakness 2:  The descriptions of some equations are insufficient; specifically, the temperature parameters in Equations (5) and (6) lack clarity regarding whether they are shared or separately optimized.**
>
> **A2:** Thank you for your comment. The temperature parameters in Equations (5) and (6) are the same and shared for both inter-view and intra-view losses. We wii clarify this in the revised manuscript to avoid confusion.
>
> **Weakness 3:   There are formatting inconsistencies, including mixed italic and upright subscripts in Equations (5) and (6), as well as inconsistent table header styles.**
>
> **A3:** Thank you for pointing this out. We will correct the formatting inconsistencies in Equations (5) and (6) and standardize the table headers in the revised manuscript.
>
> **Question 1:  The definitions and roles of inter-view and intra-view contrastive losses are not clearly articulated. It remains unclear how each specifically contributes to cross-view consistency and within-view discriminability.**
>
> **A4:** Thank you for your valuable comment. We will clarify the definitions and roles of the inter-view and intra-view contrastive losses in the revised manuscript. Specifically, the inter-view loss enforces alignment of the same instance across different views to ensure cross-view consistency, while the intra-view loss enhances discriminability within each view by leveraging hypergraph structure to prevent over-smoothing. This distinction will be clearly articulated to better demonstrate how each loss contributes to the overall representation learning.
>
> **Question 2: Equation (7) lacks a detailed explanation of the construction of the hypergraph Laplacian matrix and the mechanism behind the hyper-Laplacian regularization term.**
>
> **A5:**  Thank you for your valuable comment. A hypergraph $\mathcal{G} = (\mathcal{V}, \mathcal{E})$ is a generalized graph structure modeling high-order relationships, where $\mathcal{V}$ is the set of vertices and $\mathcal{E}$ the set of hyperedges. The hyperedges are represented by an incidence matrix $\mathbf{H} \in \\{0,1\\}^{|\mathcal{V}| \times |\mathcal{E}|}$, where $h(v,e)$ if vertex $v$ belongs to hyperedge $e$, and 0 otherwise. Each hyperedge is associated with a non-negative weight encoded in a diagonal matrix  $\mathbf{W}$. The degrees of vertices and hyperedges are defined as $d(v) = \sum_{e \in \mathcal{E}} w(e) h(v, e)$ and $d(e) = \sum_{v \in \mathcal{V}} h(v, e)$, which correspond to diagonal matrices $\mathbf{D}_v$ and $\mathbf{D}_e$, respectively. Based on these definitions, the normalized hypergraph Laplacian matrix is formulated as $\mathbf{L}_H = \mathbf{D}_v - \mathbf{H} \mathbf{W} \mathbf{D}_e^{-1} \mathbf{H}^\top$, capturing the high-order structure of the data. The hyper-Laplacian regularization term $\mathrm{tr}(\widetilde{\mathbf{P}}^{v} \mathbf{L}_H^v (\widetilde{\mathbf{P}}^v)^{\top})$ encourages smoothness of the learned representations over the hypergraph, preserving intrinsic local geometry and enhancing the model’s capacity to capture complex structural information. We will further elaborate this construction and its mechanism in the revised manuscript.
>
> **Question 3: Figure 2 indicates that the proposed method exhibits stable performance on some datasets but fluctuates significantly on others. The manuscript should discuss potential reasons for this variation in robustness.**
>
> **A6:**  The observed variation in performance stability across different datasets may be attributed to differences in data characteristics such as sample size, feature heterogeneity, and noise levels. For datasets with more complex structures or higher noise, the model may experience greater fluctuations during training. We will include a detailed discussion of these factors and their impact on robustness in the revised manuscript.
>
> **Question 4: While hypergraph construction is intended to capture high-order relations to enhance representation learning, the underlying mechanism is not sufficiently explained and would benefit from a more in-depth discussion.**
>
> **A7:**  In our method, hypergraphs are used to model high-order relationships by connecting multiple related samples through hyperedges, which allows capturing complex structural information beyond pairwise connections. This facilitates more expressive feature learning by encouraging smoothness and consistency among groups of related nodes, thus enhancing the quality of learned representations. We will expand the discussion of this mechanism in the revised manuscript to clarify its role and benefits in representation learning.
>
> **Limitations: A potential limitation of hypergraph modeling is the increased computational complexity associated with capturing high-order relationships. This aspect should be explicitly acknowledged and discussed in the manuscript.**
>
> **A8:** Thank you for your insightful comment. We acknowledge that hypergraph modeling, while effective at capturing high-order relationships, does introduce increased computational complexity compared to traditional pairwise graph methods. This complexity arises from the construction and processing of hyperedges that connect multiple nodes simultaneously. In the revised manuscript, we will provide a more detailed discussion on this point.

---

### Official Review · Reviewer_mwWy · 2025-06-30

**Clarity:** 3
**Significance:** 3
**Originality:** 4
**Rating:** 5
**Confidence:** 4

**Summary:**

In this work, the authors propose a hypergraph-enhanced contrastive learning method for multi-view clustering. The method leverages hypergraph structures and a weight-sharing hypergraph neural network to capture high-order data correlations. In addition, it introduces an inter-view sample discriminability enhancement strategy, which effectively alleviates the issue of weak discriminability caused by information propagation in hypergraph neural networks. Extensive experiments validate the feasibility and effectiveness of the proposed model.

**Questions:**

Please see Weaknesses.

**Ethical Concerns:**

["NO or VERY MINOR ethics concerns only"]

**Limitations:**

This work is a fundamental study in deep clustering and does not exhibit obvious limitations.

**Quality:**

3

**Strengths And Weaknesses:**

Strengths：
1. Introducing hypergraph neural networks into multi-view clustering effectively captures high-order data correlations. This idea is highly innovative.

2. The simultaneous use of inter-view and intra-view contrastive losses significantly enhances the discriminability of learned sample representations, which is highly beneficial for clustering tasks.

3.  A series of experiments, including ablation studies, thoroughly validate the superior clustering performance of the proposed model.

Weaknesses：

1. The detailed process of hypergraph construction is missing.

2. In Equation (9), there seems to be a typographical error in the size of the variable P.

3. In Figure 2, the font size of the axis labels and tick marks is too small.

4. The manuscript lacks convergence experiments.

5. Compared with pre-training followed by fine-tuning, how does the clustering performance of the jointly optimized approach compare?

---

> ### Author Rebuttal · Authors · 2025-07-30
>
> **Weakness 1：The detailed process of hypergraph construction is missing.**
>
> **A1:**  Thank you for your constructive comment. To elaborate, the hypergraph construction in our method is based on a k-nearest neighbors (k-NN) strategy tailored for multi-view data. Specifically, for each view, we compute the pairwise Euclidean distances between all samples to capture local geometric relationships. For each sample, we then identify its top-k nearest neighbors and form a hyperedge that connects this sample with those neighbors. This results in a set of hyperedges that represent high-order relations beyond simple pairwise connections. By leveraging these hyperedges, the hypergraph can effectively encode complex structural and contextual information inherent in each view. This design enables our model to better capture local manifold structures and preserve intrinsic data geometry, which are critical for robust and discriminative representation learning. We will provide a detailed description of this process in the revised manuscript to clarify the construction and its rationale.
>
> **Weakness 2：In Equation (9), there seems to be a typographical error in the size of the variable P.**
>
> **A2:** Thank you for pointing out this issue. We have carefully reviewed Equation (9) and confirmed the typographical error regarding the size of the variable P. The correct dimension should be  ${\mathbb{R}}^{N \times \sum\limits_{v=1}\limits^M d_{v}}$, where $d_{v}$ denotes the dimension of $\mathbf{P}^v$. We will correct this in the revised manuscript to ensure clarity and accuracy.
>
> **Weakness 3: In Figure 2, the font size of the axis labels and tick marks is too small.**
>
> **A4:** Thank you for your detailed comment. We will address this issue in the revised manuscript by enlarging the axis labels and tick marks to improve readability.
>
> **Weakness 4:  The manuscript lacks convergence experiments.**
>
> **A5**  Thank you for pointing this out. To address this concern, we will include convergence experiments in the revised manuscript to illustrate the training stability and efficiency of our proposed method.
>
> **Weakness 5:  Compared with pre-training followed by fine-tuning, how does the clustering performance of the jointly optimized approach compare?**
>
> **A5:** Thank you for this insightful comment.
> Our framework employs a two-stage optimization, pre-training followed by fine-tuning to balance view-specific representation learning and view-consistent information capture. During the pre-training process, view-specific autoencoders are trained to preserve view-specific information and obtain representative view-specific embeddings. The embeddings serve as inputs for hypergraph construction and hypergraph encoder. During the fine-tuning process, we aim to capture the view consistent information from the view-specific embeddings through contrastive learning strategy. To clarify, in this stage, the view-specific autoencoder models are frozen.
>
> Compared with joint optimization, our two-stage approach offers **explicit consistency extraction**. It enables clear separation of view-consistent information through distinct optimization phases. In contrast, jointly optimization would require handling dynamic hypergraph structures, as continuously updated embeddings from the autoencoders would modify hyperedge connectivity during training. We will further elaborate on this point in the revised manuscript.

---

> > ### Comment · Reviewer_mwWy · 2025-08-09
> >
> > Your responses have addressed all of my questions, and your work is highly constructive; therefore, I maintain my score.

---

### Official Review · Reviewer_Htnz · 2025-07-03

**Clarity:** 2
**Significance:** 2
**Originality:** 1
**Rating:** 2
**Confidence:** 4

**Summary:**

This paper proposes HOPER, a novel multi-view clustering framework that integrates hypergraph-enhanced contrastive learning with hypergraph Laplacian regularization to learn discriminative feature representations.

**Questions:**

1, Hypergraph Contrastive Learning has been proposed in: "Multi-view Mixed Attention for Contrastive Learning on Hypergraphs. In Proceedings of the 47th International ACM SIGIR Conference on Research and Development in Information Retrieval", "Multi-view Hypergraph Adaptive Contrastive Learning. In Pacific-Asia Conference on Knowledge Discovery and Data Mining", "Multi-view hypergraph contrastive policy learning for conversational recommendation. In Proceedings of the 46th international ACM SIGIR conference on research and development in information retrieval", "Cross-view graph contrastive learning with hypergraph. Information Fusion, 99, 101867."
The authors should clarify the differences between their method and other methods.
2. Hyper-Laplacian Regularization has also been proposed in: "Hyper-Laplacian regularized multi-view clustering with exclusive L21 regularization and tensor log-determinant minimization approach. ACM Transactions on Intelligent Systems and Technology, 14(3), 1-29", "Hyper-Laplacian Regularized Concept Factorization in Low-Rank Tensor Space for Multi-View Clustering. IEEE Transactions on Emerging Topics in Computational Intelligence", "Hyper-Laplacian regularized multi-view subspace clustering with low-rank tensor constraint. Neural Networks, 125, 214-223", "Hyper-Laplacian regularized nonconvex low-rank representation for multi-view subspace clustering. IEEE Transactions on Signal and Information Processing over Networks, 8, 376-388."
 The authors should also clarify the differences between their method and other methods.
3. In summary, the two main contributions have been proposed in previous methods. The authors should clarify that their method is not a simple combination of existing methods. Moreover, these methods should also be compared in experiments.

**Ethical Concerns:**

["NO or VERY MINOR ethics concerns only"]

**Limitations:**

1. The novelty is limited: Hypergraph Contrastive Learning and Hyper-Laplacian Regularization have all been proposed in previous studies.
2. The comparisons have not included previous methods using Hypergraph Contrastive Learning and Hyper-Laplacian Regularization.

**Quality:**

2

**Strengths And Weaknesses:**

Strengths：
1. The structure of this paper is great.
2. The experiments are comprehensive.
3. The motivation is clear.

Weaknesses:
1. The novelty is limited: Hypergraph Contrastive Learning and Hyper-Laplacian Regularization have all been proposed in previous studies.
2. Figure 2 is less informative.

---

> ### Author Rebuttal · Authors · 2025-07-30
>
> **Weakness 1: The novelty is limited: Hypergraph Contrastive Learning and Hyper-Laplacian Regularization have all been proposed in previous studies.**
>
> **A1:** Thank you for your valuable comment. While it is true that hypergraph contrastive learning and hyper-Laplacian regularization have been previously studied, we respectfully emphasize that our method is substantially different in both application scope and technical integration, and is not a straightforward combination of existing techniques.
>
> **First**, the application domains are different. Existing Hypergraph Contrastive Learning methods have predominantly focused on single-view node classification tasks. In contrast, our method is designed for multi-view unsupervised clustering, which poses distinct challenges and requires different methodological considerations.
>
> **Second**, although some prior works have incorporated Hyper-Laplacian Regularization into multi-view clustering, these are mostly shallow models that apply Laplacian constraints on subspace representations to preserve local structures. In contrast, our work is, to the best of our knowledge, the first to introduce Hyper-Laplacian Regularization into a deep learning-based multi-view clustering framework. Rather than regularizing subspace representations, we apply the regularization directly on the learned feature embeddings to enhance their discriminability, which in turn improves clustering performance.
>
> **In summary**, although related concepts have been explored previously, our method is novel in both its application domain and technical design. Specifically, we are, to the best of our knowledge, the first to integrate hypergraph neural networks into the multi-view clustering setting, which represents a meaningful advancement in this area.
>
> **Weakness 2: Figure 2 is less informative.**
>
> **A2:** Thank you for your comment. In the revised manuscript, we will improve Figure 2 by adding clearer annotations and visual enhancements to better convey the model architecture and overall workflow.
>
> **Question 1: Hypergraph Contrastive Learning has been proposed in [1][2][3][4], The authors should clarify the differences between their method and other methods.**
>
> **A3:** Thank you for your insightful question. While References [1]–[4] have indeed applied Hypergraph Contrastive Learning, our method differs from them in several fundamental aspects.
>
> **First**, the **application scenarios** are different. Although some of these works (e.g., [1], [2], [4]) mention “multi-view” in their titles, they actually operate on single-view data. Specifically, their so-called "multi-view" refers to augmented versions of the same view, whereas in our work, "multi-view" denotes heterogeneous features extracted from the same instance, representing truly distinct views. To further explain this difference from information theory, the multi-view data $\mathbf{X}^1$ and $\mathbf{X}^2$ in our work are extracted from $p(x_1,y)$ and $p(x_2,y)$. The multi-view data in references [1],[2],[4] are genereated from $p(x,y)$ by adding additional information randomly or user-defined data pattern.
>
> **Second**, the tasks being addressed are not the same. References [1], [2], and [4] focus on node classification tasks, while [3] deals with recommendation. In contrast, our method is designed for unsupervised multi-view clustering, which poses different challenges and objectives.
>
> **Third**, our contrastive learning mechanism is also different. We propose a novel dual contrastive learning strategy that performs both cross-view and within-view contrast. By integrating this dual mechanism, our model not only enforces global alignment across views but also enhances local discriminability within each view, leading to more robust and semantically meaningful representations.
>
> **In summary, although prior works have explored Hypergraph Contrastive Learning, they differ significantly from ours in terms of the data setting, task, and methodological design.**
>
> **Question 2: Hyper-Laplacian Regularization has also been proposed in [5][6][7][8]. The authors should also clarify the differences between their method and other methods**
>
> Thank you for your insightful comment. While it is true that Hyper-Laplacian Regularization has been utilized in previous works [5–8], our method is substantially different from these approaches in several key aspects:
>
> **Modeling paradigm:** Prior works [5–8] are based on shallow subspace clustering models. They typically use self-representation techniques to construct subspace representations, followed by spectral clustering. In contrast, our method adopts a deep learning-based framework that directly learns discriminative feature embeddings tailored for clustering, offering better scalability and representation power.
>
> **Target of regularization:** In [5–8], Hyper-Laplacian Regularization is applied to the subspace representation matrices to preserve local geometric structures within the learned subspaces. In our approach, the regularization is imposed on the latent feature embeddings generated by the deep network. This allows the model to explicitly encourage local smoothness and structure preservation in the learned feature space, which is more effective for clustering tasks.
>
> **Overall framework:** Our model integrates hypergraph neural networks, a dual hypergraph contrastive learning mechanism, and Hyper-Laplacian Regularization into a unified deep framework. This holistic design enables the model to capture both high-order correlations across views and local geometric consistency within each view, significantly improving clustering performance.
>
> **In summary**, although our work builds upon the idea of Hyper-Laplacian Regularization, we adopt a fundamentally different model structure and apply the regularization in a novel context. To the best of our knowledge, this is the first deep multi-view clustering framework to incorporate Hyper-Laplacian Regularization at the feature level. We will further clarify these differences in the revised manuscript.
>
> **Question 3:  In summary, the two main contributions have been proposed in previous methods. The authors should clarify that their method is not a simple combination of existing methods. Moreover, these methods should also be compared in experiments.**
>
> A5:  As detailed in our responses to **Q1** and **Q2**, although Hypergraph Contrastive Learning [1–4] and Hyper-Laplacian Regularization [5–8] have been studied in previous literature, our method is not a straightforward combination of existing techniques. Instead, it introduces several key innovations that distinguish it fundamentally from prior approaches:
>
> **Distinct application scenario and task:** Prior works mostly focus on single-view data or different tasks such as node classification or dialogue recommendation. In contrast, our method is specifically designed for unsupervised multi-view clustering, where each view represents truly heterogeneous feature sets of the same instance. This task poses unique challenges and requires tailored model designs.
>
> **Novel deep learning framework integration:** Existing Hyper-Laplacian Regularization methods [5–8] are shallow models applied to subspace representations, whereas we integrate hypergraph neural networks, a dual hypergraph contrastive learning mechanism, and Hyper-Laplacian Regularization into a unified deep architecture. This joint design enables learning of discriminative latent feature embeddings with strong local and global structural consistency.
>
> **Innovative contrastive learning strategy:** Our dual contrastive learning approach simultaneously enforces global alignment across views and local discriminability within views, which is not explored in prior works and is crucial for robust multi-view representation learning.
>
> **In summary, our work presents a fundamentally different and novel solution for deep multi-view clustering by innovatively integrating and extending existing techniques within a new application and model paradigm.**
>
> Regarding the suggestion to include experimental comparisons with these prior methods, we fully acknowledge the importance of empirical validation. However, most of the referenced methods address different research problems or adopt substantially different modeling paradigms. For example, [1], [2], and [4] focus on single-view node classification, [3] targets dialogue recommendation, and [5]–[8] are shallow subspace clustering models. As such, direct quantitative comparisons may not be meaningful or fair. Nevertheless, we will further highlight these distinctions in the revised manuscript and include detailed discussions to help clarify the positioning of our work within the broader research landscape.
>
> [1] Multi-view Mixed Attention for Contrastive Learning on Hypergraphs. ACM SIGIR Conference on Research and Development in Information Retrieval.
>
> [2] Multi-view Hypergraph Adaptive Contrastive Learning. PAKDD.
>
> [3] Multi-view hypergraph contrastive policy learning for conversational recommendation. ACM SIGIR conference on research and development in information retrieval.
>
> [4] Cross-view graph contrastive learning with hypergraph. Information Fusion, 99, 101867.
>
> [5] Hyper-Laplacian regularized multi-view clustering with exclusive L21 regularization and tensor log-determinant minimization approach. ACM TIST, 14(3), 1-29.
>
> [6] Hyper-Laplacian Regularized Concept Factorization in Low-Rank Tensor Space for Multi-View Clustering. IEEE Transactions on Emerging Topics in Computational Intelligence.
>
> [7] Hyper-Laplacian regularized multi-view subspace clustering with low-rank tensor constraint. Neural Networks, 125, 214-223".
>
> [8] Hyper-Laplacian regularized nonconvex low-rank representation for multi-view subspace clustering. IEEE Transactions on Signal and Information Processing over Networks, 8, 376-388.

---

> > ### Comment · Reviewer_Htnz · 2025-08-02
> > **Response to the rebuttal**
> >
> > After I read the rebuttal, my concerns still remain.
> >
> > 1. It is claimed by the authors that "Existing Hypergraph Contrastive Learning methods have predominantly focused on single-view node classification tasks". I still find lots of papers have studied this:
> > Yu, Z., Fu, L., Chen, Y., Cai, Z., & Chao, G. (2024). Hyper-Laplacian regularized concept factorization in low-rank tensor space for multi-view clustering. IEEE Transactions on Emerging Topics in Computational Intelligence.
> > Wang, S., Chen, Y., Zhang, L., Cen, Y., & Voronin, V. (2022). Hyper-Laplacian regularized nonconvex low-rank representation for multi-view subspace clustering. IEEE Transactions on Signal and Information Processing over Networks, 8, 376-388.
> > Xie, Y., Zhang, W., Qu, Y., Dai, L., & Tao, D. (2018). Hyper-Laplacian regularized multilinear multiview self-representations for clustering and semisupervised learning. IEEE transactions on cybernetics, 50(2), 572-586.
> > Song, P., Zhou, S., Mu, J., Duan, M., Yu, Y., & Zheng, W. (2024). Clean affinity matrix induced hyper-Laplacian regularization for unsupervised multi-view feature selection. Information Sciences, 682, 121276.
> >
> > 2. It is claimed by the authors that "the first to introduce Hyper-Laplacian Regularization into a deep learning-based multi-view clustering framework". I also find lots of related papers:
> > Xie, Y., Zhang, W., Qu, Y., Dai, L., & Tao, D. (2018). Hyper-Laplacian regularized multilinear multiview self-representations for clustering and semisupervised learning. IEEE transactions on cybernetics, 50(2), 572-586.
> > Song, P., Zhou, S., Mu, J., Duan, M., Yu, Y., & Zheng, W. (2024). Clean affinity matrix induced hyper-Laplacian regularization for unsupervised multi-view feature selection. Information Sciences, 682, 121276.
> > Yu, X., Liu, H., Zhang, Y., Gao, Y., & Zhang, C. (2025). Robust multi-view clustering with hyper-Laplacian regularization. Information Sciences, 694, 121718.
> > Che, H., Li, C., Leung, M. F., Ouyang, D., Dai, X., & Wen, S. (2024). Robust hypergraph regularized deep non-negative matrix factorization for multi-view clustering. IEEE Transactions on Emerging Topics in Computational Intelligence.
> > Huang, H., Zhou, G., Liang, N., Zhao, Q., & Xie, S. (2022). Diverse deep matrix factorization with hypergraph regularization for multi-view data representation. IEEE/CAA Journal of Automatica Sinica, 10(11), 2154-2167.
> >
> > I truly hope more interesting problems to be proposed in multi-view clustering rather than combining existing techniques to form a new method.

---

> ### Author Response · Authors · 2025-08-03
>
> We sincerely appreciate the reviewer’s detailed comments and for citing several relevant works in the field.
>
> **Clarification on Hypergraph Contrastive Learning**
> Regarding our statement in the rebuttal that *“existing hypergraph contrastive learning methods primarily focus on single-view node classification tasks,”* we would like to further clarify that this statement specifically refers to **Hypergraph Contrastive Learning (HCL)** methods, rather than to all works that utilize hypergraphs in general. While the cited works [1]–[4] employ hypergraph modeling or hyper-Laplacian regularization in multi-view contexts, none of them leverage contrastive learning strategies—particularly not those built upon hypergraph structures. Therefore, within the specific context of *hypergraph contrastive learning*, we believe our original statement is accurate and valid.
>
> **On the Use of Hyper-Laplacian Regularization in Deep Multi-View Clustering**
> Concerning our claim that we are the *“first to introduce hyper-Laplacian regularization into a deep learning-based multi-view clustering framework,”* we understand that the reviewer has pointed out several related works that also adopt hyper-Laplacian regularization. However, it is important to emphasize that these methods are based on **shallow learning paradigms**, such as matrix/tensor factorization or graph self-representation techniques [3]–[7], and do not incorporate deep neural network architectures capable of learning hierarchical and discriminative representations within a unified framework. In contrast, to the best of our knowledge, our work is the **first to integrate hypergraph neural networks, a dual hypergraph contrastive learning strategy, and hyper-Laplacian regularization** within a unified deep learning framework to enhance multi-view clustering performance. Thus, our “first-of-its-kind” claim is made specifically in the context of **deep learning-based multi-view clustering**, which we believe is a fair and appropriate characterization.
>
> **On the Reviewer’s Suggestion Regarding Innovation**
> We also fully acknowledge the reviewer’s comment that *“posing more novel problems is preferable to merely combining existing techniques.”* In this regard, we would like to emphasize that our method is **not** a simple aggregation of existing techniques, but a purposeful and novel integration motivated by the unique challenges of deep multi-view clustering. As detailed in our initial rebuttal, our key contributions include:
>
> - **Unique task formulation**: Unlike existing HCL methods focused on single-view tasks (e.g., node classification), our method addresses **unsupervised multi-view clustering**, where each view offers heterogeneous features—posing greater modeling challenges.
>
> - **Deep framework integration**: While prior works apply hyper-Laplacian regularization in shallow settings, we are the first to integrate **hypergraph neural networks, a dual hypergraph contrastive learning strategy, and hyper-Laplacian regularization** into a unified deep architecture for structure-aware representation learning.
>
> - **Novel contrastive learning design**: Our **dual contrastive strategy** jointly promotes cross-view alignment and within-view discriminability—an approach not explored in prior multi-view clustering literature and essential for effective representation learning.
>
> **In conclusion,** while leveraging existing techniques is common, our work goes beyond mere combination by addressing key challenges in deep multi-view clustering through a unified and innovative framework. This approach extends prior methods in both design and formulation, offering clear novelty and practical value to the field.
>
> [1]  Yu, Z, et al. (2024). Hyper-Laplacian regularized concept factorization in low-rank tensor space for multi-view clustering. IEEE Transactions on Emerging Topics in Computational Intelligence.
>
> [2] Wang, S, et al. (2022). Hyper-Laplacian regularized nonconvex low-rank representation for multi-view subspace clustering. IEEE Transactions on Signal and Information Processing over Networks, 8, 376-388.
>
> [3] Xie, Y, et al. (2018). Hyper-Laplacian regularized multilinear multiview self-representations for clustering and semisupervised learning. IEEE TCYB, 50(2), 572-586.
>
> [4] Song, P, et al. (2024). Clean affinity matrix induced hyper-Laplacian regularization for unsupervised multi-view feature selection. Information Sciences, 682, 121276.
>
> [5] Yu, X, et al. (2025). Robust multi-view clustering with hyper-Laplacian regularization. Information Sciences, 694, 121718.
>
> [6] Che, H, et al. (2024). Robust hypergraph regularized deep non-negative matrix factorization for multi-view clustering. IEEE Transactions on Emerging Topics in Computational Intelligence.
>
> [7] Huang, H, et la. (2022). Diverse deep matrix factorization with hypergraph regularization for multi-view data representation. IEEE/CAA Journal of Automatica Sinica, 10(11), 2154-2167.

---

> ### Author Response · Authors · 2025-08-06
>
> Dear Reviewer Htnz,
>
> Thank you again for your thoughtful comments and for pointing out relevant prior works. In our rebuttal, we have carefully addressed the distinctions between our method and the ones you mentioned, and we further clarified that—while leveraging existing techniques is common—our work goes beyond a mere combination by tackling key challenges in deep multi-view clustering through a unified and innovative framework. This extends prior approaches in both design and formulation, offering clear novelty and practical relevance.
>
> We hope our clarification has resolved your concerns. If there are any remaining questions or uncertainties, we would be grateful for any further feedback you might share.
>
> Sincerely,
>
> The Authors

---

### Decision · Program_Chairs · 2025-09-17

**Decision:**

Accept (poster)

**Comment:**

This paper presents a new hypergraph-enhanced contrastive learning framework for multi-view clustering, which effectively integrates hypergraph neural networks with dual contrastive learning and hyper-Laplacian regularization to capture high-order sample dependencies. While one reviewer raised concerns about novelty, the other three reviewers recognized its innovation, where the integration of these components into a unified framework, along with inter-view discriminability enhancement, demonstrates practical innovation and empirical effectiveness. Considering the method’s superior performance across multiple benchmarks and its novel use of view-specific hypergraphs address key limitations in existing multi-view clustering approaches, the paper is recommended for acceptance.